# Towards the Fabrication of High-Aspect-Ratio Silicon Gratings by Deep Reactive Ion Etching

**DOI:** 10.3390/mi11090864

**Published:** 2020-09-18

**Authors:** Zhitian Shi, Konstantins Jefimovs, Lucia Romano, Marco Stampanoni

**Affiliations:** 1Paul Scherrer Institut, 5232 Villigen PSI, Switzerland; konstantins.jefimovs@psi.ch (K.J.); lucia.romano@psi.ch (L.R.); stampanoni@biomed.ee.ethz.ch (M.S.); 2Department of Information Technology and Electrical Engineering, ETH Zürich, 8092 Zürich, Switzerland; 3Department of Physics and CNR-IMM-University of Catania, 64 via S. Sofia, 95123 Catania, Italy

**Keywords:** X-ray grating interferometry, aspect ratio dependent etching (ARDE), X-ray energy range

## Abstract

The key optical components of X-ray grating interferometry are gratings, whose profile requirements play the most critical role in acquiring high quality images. The difficulty of etching grating lines with high aspect ratios when the pitch is in the range of a few micrometers has greatly limited imaging applications based on X-ray grating interferometry. A high etching rate with low aspect ratio dependence is crucial for higher X-ray energy applications and good profile control by deep reactive ion etching of grating patterns. To achieve this goal, a modified Coburn–Winters model was applied in order to study the influence of key etching parameters, such as chamber pressure and etching power. The recipe for deep reactive ion etching was carefully fine-tuned based on the experimental results. Silicon gratings with an area of 70 × 70 mm^2^, pitch size of 1.2 and 2 μm were fabricated using the optimized process with aspect ratio α of ~67 and 77, respectively.

## 1. Introduction

X-ray grating interferometry is a powerful method for non-destructive X-ray phase contrast imaging of weakly absorbing low atomic number materials which has a high impact in multiple fields, such as medical diagnostics, archeology, material science, etc. [1,2,3,4]. One of the prominent features of X-ray grating interferometric systems is that the essential optical components of the experimental setups are gratings with high aspect ratio (HAR) [5]. Two different types of grating are involved in X-ray grating interferometry: phase gratings for creating the interference pattern (G1); absorption gratings for generating coherent light from conventional X-ray tubes (G0) and for analyzing the fine fringes in front of the detector (G2) [1]. HAR patterns are usually created as templates and then filled by high X-ray absorbing material, such as gold. The desire to increase the aspect ratio comes from the developing trend of X-ray grating interferometry: the smaller the grating period, the higher the phase contrast sensitivity [6], and the thicker the grating height, the higher the applicable X-ray energy range [7].

For the fabrication of HAR gratings, especially G0 and G2 gratings, researchers have several mature technologies to choose from: LIGA (Lithographie, Galvanik und Abformung) [8,9], anisotropic wet-etching [10], metal assisted chemical etching (MacEtch) [11,12,13] and deep reactive ion etching (DRIE) [9,11,14,15]. The LIGA process relies highly on the availability of a synchrotron source; wet-etching gives very good sidewall smoothness but has poor etching depth uniformity and is very sensitive to precise crystallographic alignment; MacEtch could achieve an extremely high aspect ratio, but still it has limited reliability [13]. DRIE is one of the most promising, industrially relevant methods for fabricating HAR gratings in silicon with pitch in the range of a few micrometers on large areas and scalable to mass production. The most commonly adapted time division multiplex method for deep silicon etching was developed and patented by F. Laermer and A. Schilp, which is referred to as the Bosch etching method [16]. Teflon-like polymers are generated and attached on the sidewall during the deposition sub-steps to enhance anisotropic etching. However, the major challenges are related to macro- and micro-loading effects. It has been observed that the etch rate in reactive ion etching depends on the mask-free area (RIE-lag), with smaller etch rates in smaller features [17,18], or the etching is aspect ratio dependent (ARDE), meaning that the etch rate decreases as a function of etch time for a given line width. These effects become relevant and induce the decreasing of the etching rate when fabricating large area gratings with periods in the micrometer range. Knudsen’s transport of etchants and byproducts dominates the etch rate when the mean free path of the particles becomes larger than half of the grating period [19]. The plasma etching is modeled in terms of three basic steps: adsorption, product formation and product desorption by Coburn and Winters [20]. Clausing [21] describes the transmission probability as a shape-dependent-only quantity, which was determined by the geometry of the channel. 

The Bosch etching method is mostly used for the fabrication of microelectromechanical systems (MEMS), which contain features that have several different pitches and pattern densities. Therefore, the loading and RIE-lag effects are usually studied either with discrete lines containing different pitch sizes and densities or a special ‘nested broken rings’ pattern. For grating fabrication, the pattern on the silicon substrate covers more than 60% of the entire wafer to ensure a larger field of view in X-ray grating interferometry setups, which results in 30% of the exposed area for a grating duty cycle of 0.5, and this value will be even larger if the duty cycle of the grating is lower than 0.5. Moreover, unlike MEMS, usually only one particular critical dimension of the grating pattern has really a major impact on the etch rate, which is the period of the grating lines. Therefore, the conclusions from previous studies [17,22,23,24,25] are not fully representative of our application. The main reason for this is the fact that most of the DRIE applications like MEMS are targeted at achieving uniform depth for features with different sizes. In contrast, minimizing the feature size and reaching the highest possible aspect ratio is the main challenge in grating fabrication for X-ray imaging applications. This paper studied the impact of the most critical parameters on ARDE effects during Bosch etching by using the well-established Coburn–Winters model for the specific HAR gratings fabrication. In particular, we studied the etching of grating with a pitch size in the range between 1.2 and 10 μm at different chamber pressures and etching powers, which ensure stable conditions. 

## 2. Materials and Methods

N-type silicon wafers with diameter of 100 mm and crystal orientation of <100> were chosen as the substrates for the grating fabrication. Wafers with lower resistances (0.001–0.01 Ω·cm) are preferred to facilitate the following gold electroplating process, for filling the silicon templates with highly absorbing X-ray materials in order to create the absorption gratings [26]. Either standard photolithography (for gratings with periods larger than 2 μm) or displacement Talbot lithography (DTL) (for gratings with periods smaller than 2 μm) [15] was used for preparing the pattern of grating lines. MICROPOSIT S1800 series positive photoresists were used for standard photolithography, and SUMIRESIST PFI-88A7 positive photoresist together with AZ Barli-II bottom anti-reflective coating (BARC) were used for DTL. The exposed pattern of grating lines was developed in Megaposit MF-24A developer. The transfer into the underlying layer of chromium was done by plasma etching, with Cl_2_ and O_2_ as etchant gases. The chromium layer served as a hard mask for the following silicon DRIE process. The thickness of the chromium layer varied from 50 nm to 130 nm, depending on the target height of the grating lines. Figure 1 illustrates the used procedure of grating fabrication.

The DRIE process was realized in PlasmaLab100 ICP-type etcher from Oxford Instruments Plasma Technology, using C_4_F_8_ and SF_6_ as passivation and etching gases, at a fixed temperature of 0 °C. Scanning electron microscopy (SEM) with In-Lens detector of a Zeiss Supra VP55 was used to characterize the etched gratings in cross-section view. At the beginning of each process, the depths of the trenches are not very high, so we could assume that the etch rate is not yet affected by the ARDE effect. We therefore measured the initial etch rate in a simple way, as described below. Figure 2 shows a schematic (Figure 2a) and a SEM image of the scalloped etched profile (Figure 2b) and the measurement of the height corresponding to five etching loops.

The switching points of etching and deposition sub steps are visible as small peaks on the sidewall. The initial etch rates (*R*_0_) for each etching condition were calculated according to Equation (1):(1)$$R_{0}\;=\;\frac{H_{5\times \mathrm{scallops}}}{5\;\times \;\tau }$$
where *H*_5×scallops_ is the distance between the second peak and the seventh peak of the scalloped profile (see Figure 2a), and *τ* is the time of one etching sub-step.

The key parameters of the DRIE process listed in Table 1 are chamber pressure, radiofrequency (RF) and inductively coupled plasma (ICP) powers for etching and deposition steps. The etchant gas flows were adjusted according to the set chamber pressure in order to have a stabilized chamber pressure during the etching process.

## 3. Results

### 3.1. Modified Coburn–Winters Model

During the process of reactive ion etching, the etchant gas is ionized and transferred to the etching area, the radicals react with silicon atoms at the exposed surface, and the etching residuals are removed outside of the etched trenches. In general, three prerequisites are indispensable to achieve a high etching rate in a DRIE process: (i) sufficient amount of etchant being supplied; (ii) the ions accelerated in the sheath region should have high enough kinetic energy; (iii) the neutrals and byproducts should be efficiently pumped out of the deep trench. Especially in DRIE, the etch rate is dominated by Knudsen’s transport of etchants and byproducts when the mean free path of these particles becomes larger than half of the grating period [19]. We need a model to analyze the dropping trend of the etch rate under different etching conditions. In order to better understand how the etching parameters affect the etch rate, we studied the data by utilizing the well-established Coburn–Winters model, and Equation (2) shows the normalized instantaneous etch rate of the reactive ion etching process:(2)$$\frac{R}{R_{0}}\;=\;\frac{K}{K\;+\;S\;-\;\mathrm{KS}}$$
where *R* is the etch rate, *K* is the probability that a randomly directed etchant travels from the opening to the bottom of the deep trench, and *S* is the reaction probability of the etchant with silicon atoms.

The grating line height (*h*) is directly measured from the etched silicon wafers, and the average etch rate is calculated by the ratio between h and the total etch time (*t*), assuming a constant etch rate throughout the entire etching process. However, the instantaneous etch rate (*R*) is the gradient of the height vs. the time. According to Clausing [21], for slit-like structures, an approximation (Equation (3)) can be applied to represent the transmission probability K, if the length of the slit is much larger than the width and the height of the slit is much larger than the width (aspect ratio *α* >> 1):(3)$$K\;=\;\frac{1}{\alpha }\ln\left(\alpha \right)\;$$
where *α* is the aspect ratio of the slit-like structure. Figure 3 reports the comparison of the values calculated with Equation (3) and those from O’Hanlon et al. [27]. We found that the difference was below 4.4% for aspect ratio α larger than 10.

Trenches between grating lines are slit-like structures, since both the length and the height of the grating lines are much larger than half of the grating period. Besides, all the gratings that we fabricated for the X-ray grating interferometry have aspect ratios α larger than 10, meaning the approximated K (Equation (3)) is suitable for our application.

By replacing the transmission probability K with Equation (3), the etching rate of the Coburn–Winters model can be expressed by Equation (4): (4)$$R\;=\;\frac{\frac{\ln(\alpha )}{\alpha }R_{0}}{\frac{\ln(\alpha )}{\alpha }+\;S\;-\;\frac{\mathrm{Sln}(\alpha )}{\alpha }}.$$

*R* being the gradient of height vs. time, Equation (5) is obtained by integrating Equation (4) and is used to fit the experimental data: (5)$$(1\;-\;S)h+\frac{{\mathrm{Sh}}^{2}\ln\left(\frac{2h}{p}\right)}{p}\;-\;\frac{{\mathrm{Sh}}^{2}}{2p}{\;=\;R}_{0}t\;.$$

The grating height (*h*) and the total etch time (*t*) are retrieved from real measurements, the grating period (*p*) is a known value, the starting etch rate *R*_0_ has been measured according to Equation (1) with the method introduced above (see values in Table 1). It is relevant to note that, since an aspect ratio α higher than 10 was presumed, this model only applies for slit-like structures with aspect ratio α higher than 10. 

### 3.2. Fitting of the Experimental Data

The modified model according to Equation (5) fits the experimental data of height vs. time. Figure 4 shows the experimental data and the modeling for three different sets of gratings with periods of 1.2, 3 and 5.25 µm, respectively.

Table 2 reports the *S* values fitted from the experimental data with the modified Coburn–Winters model (Equation (5)). As a general trend, for etching conditions under the same RF/ICP power, the reaction probability *S* values are relatively higher under higher chamber pressures, since a larger amount of etchants leads to a higher etchant concentration. With the same process pressure, the reaction probability S decreases when lower RF/ICP powers are applied (see, for example, the data of Table 2 for Recipe No. 2 vs. 3 and Recipe No. 5 vs. 6). This is mainly because the lower RF/ICP power naturally results in a lower DC bias, and the ions could not acquire enough kinetic energy under such conditions.

With the *S* values fitted from experimental data, the instantaneous etch rate (*R*) as a function of the aspect ratio (*α*) is calculated according to Equation (4) and plotted with solid lines in Figure 5. The ratio between the instantaneous etch rate (*R*) and the starting etch rate (*R*_0_) is called normalized etch rate and is plotted with dotted lines in Figure 5 for the three sets of gratings. Figure 5 represents the theoretical projection of etching rate with the used experimental conditions. The main limitation preventing the achievement of such a high aspect ratio (*α* > 100) is the limited hard mask thickness and the difficulty of duty cycle control over the entire etching depth.

## 4. Discussion

### 4.1. Effect of Process Pressure

The flows of etchant gases were set based on the process pressures, so that the advanced pressure control unit of the etcher could work in the optimal operational range, in order to have a stable chamber environment. The higher the pressure, the larger the amount of process gas that is injected into the etcher. Since more etchant is provided, the starting etch rate naturally increases. However, the ion angular distribution depends on the pressure, so, with increasing pressure, the ions suffer more from the geometrical shadowing effect [28,29,30]. On the other hand, it will be more difficult to pump out the neutrals and byproducts, as the mean free path of the radicals decreases. This kind of phenomenon becomes even more prominent when increasing the aspect ratio [18,24,31,32]. Although the general trend is the same for all our tests, from Figure 2 and Figure 3, we noticed that the actual etching behavior depends on the grating period. For gratings with periods of 2 μm and above, the relatively wider trench openings allow efficient byproduct removal, even at high chamber pressures; therefore, a higher chamber pressure is adapted to have a higher overall etch rate. When the period of the gratings shrinks down to the size of around 1μm, the etching process becomes more sensitive to the byproduct removal rate, and the depletion of fluorine radicals becomes less influential; hence, a lower chamber pressure is preferred to guarantee efficient Knudsen transport of ions and neutrals. 

### 4.2. Effect of Bias Power

The etcher has an ICP power supply to generate plasma from the etchant gas, while the RF power mainly contributes to accelerating the as-generated ions towards the sample surface. Both ICP and RF in combination determine the bias power during the process. The density of the generated plasma mainly depends on ICP: the higher the ICP power, the higher the concentration of etchant radicals during the etching process. Instead, the bias voltage in the sheath region increases with RF power accelerating the ions: the higher the bias, the higher kinetic energies are supplied. While the plasma density contributes more to the chemical etching component of the etching process, the bias voltage mainly governs the physical etching component [33]. In both cases, the overall etch rate—especially the starting etch rate—increases as the bias power increases. Higher bias power also helps to reduce the ‘scallop’ defects which are commonly seen in the time division multiplexed etching processes [34]. However, in real applications, higher powers are not always preferred. The major drawback of using high bias power is that the selectivity of the etching mask decreases [35]. The consumption of the etching mask not only limits the maximum reachable etching depth but also increases the probability of causing defects—the so-called ‘grass’—at the bottom of the trenches, due to the micro-masking effect of the redeposited hard mask material [36]. 

As observed from Figure 2 and Figure 3, the instantaneous etch rate drops faster in processes with high bias power. Two reasons could be the root causes of this trend. One reason is that the large voltage drop only exists in the sheath area, and the field deep inside of the trench can be considered as equipotential. Ions that participate in the etching process are only accelerated before and shortly after they enter the trench. As the aspect ratio increases, once the ions enter the deep trench, the high bias power does not affect the ions’ acceleration. Secondly, the flux of neutrals is not considerably influenced by the bias power. However, the angular distribution of ions becomes broader when a higher bias power is applied to the system [37]. This leads to a substantial decrease in the ions’ flux travelling through a HAR structure, since many ions with larger relative angles are either filtered out by the geometrical shadowing effect or deflected by the charged sidewall [38]. 

### 4.3. Optimized Etching Results

Despite all the limiting factors, processes with higher bias power still show higher overall etch rate. One possible way to implement the high bias power while avoiding the drawbacks is to apply pulsed bias power technology [39]. This would help to reduce the negative effects through charging neutralization during the ‘off’ period of the loops. The instrument used for this study does not provide this option. However, even without pulsed bias, it is still possible to optimize the etching by choosing the highest possible bias power and without overly sacrificing the mask selectivity and the ions’ flux.

We optimized the etching recipes for the grating fabrication based on the etch rate analysis with our modified model. For the fabrication of gratings with periods of 1.2 μm, the removal of residuals is quite relevant. Therefore, we chose 15 mTorr as the process pressure. We preferred a relatively higher process pressure of 25 mTorr in order to provide adequate etchant for the fabrication of gratings with periods of 2 μm. A RF/ICP power of 50 W/800 W was applied for both cases in order to have a high initial etching rate while keeping acceptable hard mask selectivity. Figure 6 shows some examples of HAR gratings with periods of 1.2, 2.0 and 9.92 μm. The gratings have a height of 40 μm (aspect ratio *α* ≈ 67), 77 μm (*α* ≈ 77) and 231 μm (*α* ≈ 47), respectively. In this experimental work, we used a 90 nm-thick layer of Cr as hard mask for the Bosch etching process, corresponding to selectivity of larger than 400:1, 850:1 and 2500:1 for periods of 1.2, 2.0 and 9.92 μm, respectively. The widening appearing at the top of the grating lines in Figure 6c reflects extra polymers which are accumulated during the process; the polymer residuals could be easily removed by conventional SF_6_/O_2_-based plasma treatment after the Bosch etching. There are slight differences in the height of the trenches if comparing the edge area with the central area on the wafer. This relates to the period of the grating: smaller the period, larger the difference (period 1.2 μm: <2.5%; period 2 μm: <1.5%). However, these differences are within the tolerance of X-ray grating interferometry applications. Besides, no significant difference was observed in duty cycle and sidewall angle shape, which have a higher impact on the X-ray grating interferometry imaging quality.

### 4.4. X-ray Energy Application Range

The grating-based interferometry was introduced for visible light in 1971 by Lohmann and Silva [40] and then translated to the X-ray domain [1,41,42,43]. The contrast is based on the formation of an intensity pattern thanks to the transmission through a grating. The pattern stems from interferences resulting from the diffraction of the beam by a phase grating G1. This so-called Talbot effect was already described by H. Talbot in 1836 for visible light [44]. The displacement of the intensity pattern due to the sample can either be recorded directly with a high resolution X-ray detector [45] or by means of an analyzer grating G2 [1]. The X-ray beams interfere downstream of G1 and an intensity pattern is produced in the plane of G2. G2 is an absorbing mask with a periodicity matching that of the interference fringes. Considering a rectangular phase grating of duty cycle 0.5, meaning a grating that induces periodical rectangular modulations of the wavefront without attenuating the beam, the amplitude of the phase modulations Δϕ is given by Equation (6) [46]:(6)Δϕ= (2π/λ)δh
where *h* designates the height of the grating profile, *λ* is the wavelength of the X-ray beam and *δ* is the refraction coefficient of the grating material. The interferometric set-up is usually designed to have a phase shift Δ*ϕ* equal to π or π/2, which corresponds to a maximum of intensity in the interference pattern. The grating height for π- or π/2- phase shift is then obtained by Equation (6) as a function of the designed wavelength [46]. 

The absorption gratings G2 have in principle no strict conditions in height. Ideally, the transmission of the grating bars is defined by Equation (7): (7)$$\frac{I}{I_{0}}=\;\exp\;(-\mu h)$$
where *μ* is the linear absorption coefficient of absorbing material at wavelength *λ* and h the height of the grating. The absorbing grating lines should block the X-ray beams as much as possible; for this reason, absorption gratings are usually fabricated in gold, which is one of the best X-ray absorbing materials and can be easily deposited by electroplating. Only 1/e of the total number of photons could travel through the gold lines of absorption grating G2 if the height of the grating is equal to 1/*μ* at the designed energy, and such gratings already provide good performance for sample imaging. However, for some applications, a transmission of even less than 1/e^2^ is preferred.

Figure 7 reports the calculated grating height as a function of X-ray energy by using Equation (6) for π-phase shifting Si gratings and Equation (7) for absorption gratings, where the Si templates are filled with Au. The attenuation length of Au is around 20 µm at 30 keV [47]; for higher X-ray energy, the data from NIST (National Institute of Standards and Technology) database [48] were used to compute the grating height of Au in Figure 7b.

The maximum depths etched for each grating period are marked with the horizontal blue lines in Figure 7, which correspondingly indicate the applicable X-ray energy ranges. The maximum Si grating heights produced in this work are marked with empty symbols and can be used directly as phase-shift gratings, as indicated in Figure 7a for phase shift of π. For π/2-phase shifting, the required Si gratings height is two times lower. Figure 7 illustrates, for example, that a π-phase shift could be introduced to an X-ray with energy of up to 31 keV directly by Si gratings with periods of 1.2 μm and up to 180 keV for gratings with period of 9.92 μm. Furthermore, the application range of phase gratings can be extended up to 100 keV, even for 1.2 μm pitch gratings if they are filled with Au. The covered energy range is two times higher in the case of π/2-phase shift.

Figure 7b indicates the application range of absorption gratings if the Si template is filled with Au (Figure 7b), for example, by electroplating. Two cases, for transmission level 1/e^2^ (solid line) and 1/e (dotted line), are shown. The actual filling of the Si gratings with Au is out of the scope of this paper. However, a few experimental data from the literature for the Au absorption gratings with periods of 1.2 μm [49], 3 μm [50], 5.25 μm [50], 6 μm [51] and 14 μm [52] are presented in Figure 7b as a benchmark. Figure 7 represents the state of the art of grating fabrication using Si DRIE technology and the prospects for future application. For example, Figure 7a indicates that the π-phase shifting gratings can be produced in silicon for X-ray energy up to 80 keV for a grating period of 3 μm. If this grating is filled with Au (see Figure 7b), it will act as absorption grating with application ranges up to 57 keV and 80–100 keV (the gap in application range at 57–80 keV is due to the K absorption edge of Au). So far, this has been demonstrated with a grating height of 85 μm (diamond in Figure 7b) by bottom-up Au filling [50]. Similarly, one can see that the reported etching results for HAR structures open the route for the fabrication of G2 gratings at energy up to 40 keV for a period above 1.2 µm and up to 52 keV and around 80–90 keV for a period above 2 µm. Application energies higher than 100 keV will be accessible with grating periods in the range of 5–10 μm. The reported Au gratings fabrication based on Si DRIE templates indicate that the aspect ratio is limited by that achieved in Si. The results reported here allow us to extend the application energy range of the absorption gratings. Gold electroplating remains the most popular method for filling the Si trenches, but other methods can also be potentially used for different absorbing materials [53,54,55].

## 5. Conclusions

The etching behavior of DRIE processes using different recipes was tested and analyzed. Based on the mechanism of the silicon DRIE process, the most commonly adapted Coburn–Winters model was modified and adapted in order to extract the trend of instantaneous etch rate from experimental measurements. The modified model well represents the experimental data; therefore, we could utilize this model for etching recipe development and etching behavior prediction. According to the periods of the gratings, the important etching parameters like chamber pressure and etching power were carefully balanced to have an optimum etch rate. So far, for gratings with periods of 1.2, 2, 3, 5.25 and 9.92 μm, the grating lines were etched into the silicon substrate to a depth of 40, 77, 103, 120 and 231 μm, resulting in aspect ratios of 67, 77, 69, 46 and 47, respectively, at a full 4-inch wafer scale. The maximum grating heights mentioned above are not maximum reachable heights (sometimes referred to as critical aspect ratio), since the etching rates are not yet limiting the process. A π-phase shift could be introduced to an X-ray energy of up to 31 keV directly by Si gratings with a period of 1.2 μm, and up to 180 keV for gratings with a period of 9.92 μm. The DRIE results reported here allow us to extend the application energy range of absorption gratings, compared to those reported to date. Further application of continuous parameter ramp during the etching, as well as more sophisticated plasma control such as pulsed bias, allows substantial increase of the achievable aspect ratios. Our future work also includes investigation and optimization of the silicon trench profile using different gold filling methods for X-ray imaging applications at small pitch and high energies (>40 keV).

## Figures and Tables

**Figure 1 micromachines-11-00864-f001:**
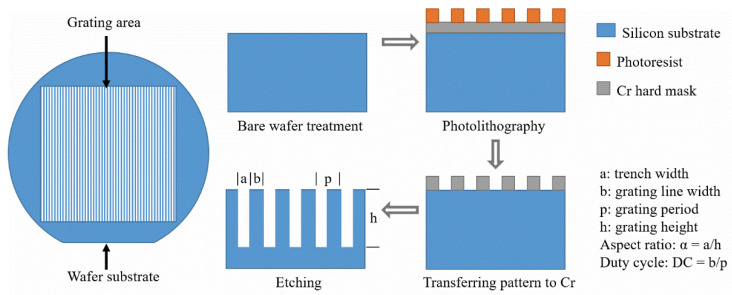
Process flow of silicon grating fabrication.

**Figure 2 micromachines-11-00864-f002:**
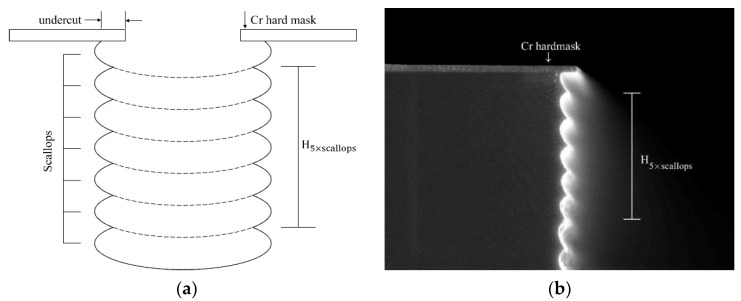
Measuring method of the initial etch rate: (**a**) schematic; (**b**) SEM image in cross-section of a real etched grating. The height of 5 scallops is used to measure the initial etch rate.

**Figure 3 micromachines-11-00864-f003:**
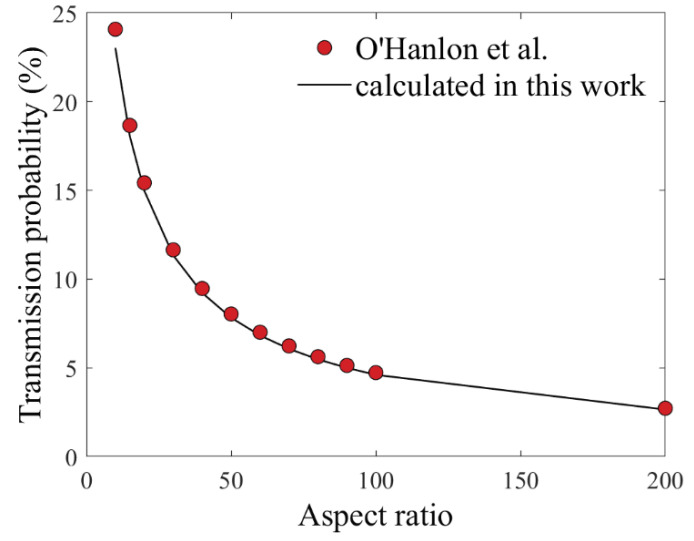
Transmission probability K calculated with Equation (3) (solid line) and data (dots) from O’Hanlon et al. Reproduced with permission from [27].

**Figure 4 micromachines-11-00864-f004:**
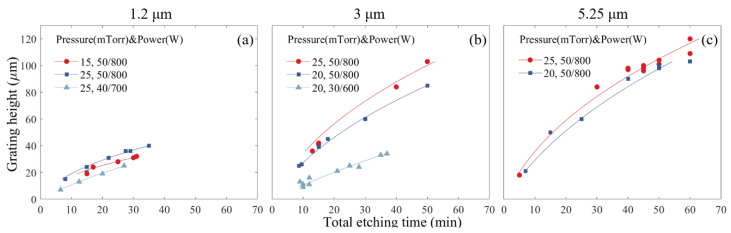
Grating height as a function of etching time for different grating periods: (**a**) 1.2 µm; (**b**) 3 µm; (**c**) 5.25 µm. The dots are raw experimental data, and the solid lines are best fits with modified Coburn–Winters model (Equation (5)).

**Figure 5 micromachines-11-00864-f005:**
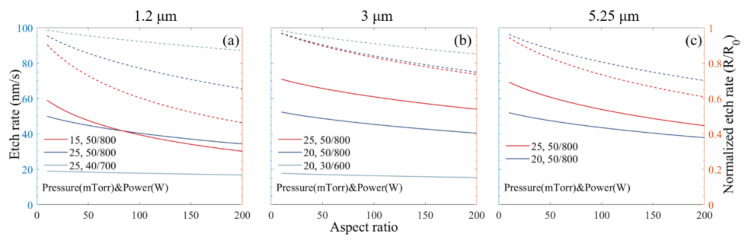
Solid lines: etch rate (*R*) as a function of aspect ratio (*α*) fitted with Equations (3)–(6); dotted lines: normalized etch rate (*R*/*R*_0_) for different grating periods: (**a**) 1.2 μm; (**b**) 3 μm; (**c**) 5.25 μm.

**Figure 6 micromachines-11-00864-f006:**
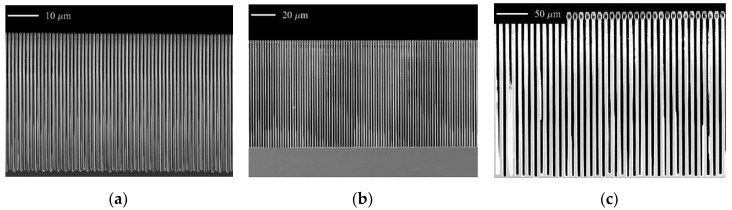
SEM images of cross-section of gratings. Grating period, height and aspect ratio: (**a**) *p* = 1.2 μm, *h* = 40 μm, *α* = 67; (**b**) *p* = 2 μm, *h* = 77 μm, *α* = 77; (**c**) *p* = 9.92 μm, *h* = 231 μm, *α* = 47.

**Figure 7 micromachines-11-00864-f007:**
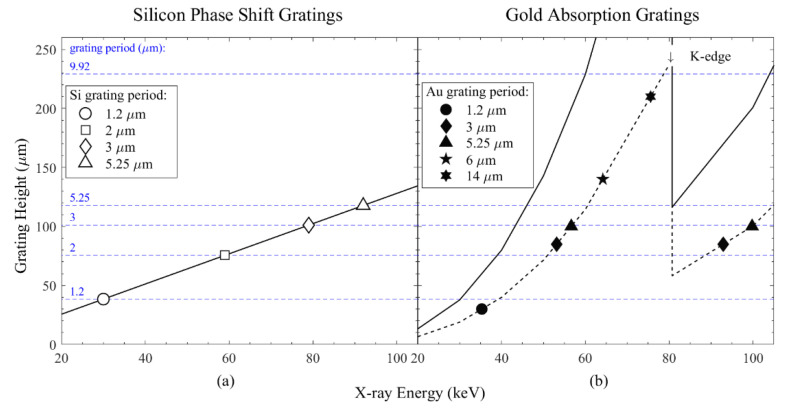
Grating height as a function of X-ray energy range of (**a**) π-phase shift Si gratings (calculated using Equation (6)) and (**b**) absorption gratings, when Si templates are filled with Au (calculated using Equation (7)). The photon intensity reduction is calculated at 1/e (dotted line) and 1/e^2^ (solid line) absorption level. The empty symbols and horizontal blue lines in (**a**) represent experimental data on Si etched grating reported in this work. The solid dots in (**b**) correspond to Si DRIE gratings filled with Au reported elsewhere (1.2 μm [49], 3 μm [50], 5.25 μm [50], 6 μm [51], 14 μm [52]). The projection of the horizontal blue lines from plot (**a**) to plot (**b**) indicates that higher energy coverage can be obtained if the Si gratings produced in this work are filled with Au. (For simplicity, the experimental data points on absorption gratings are illustrated on 1/e curve only.)

**Table 1 micromachines-11-00864-t001:** Deep reactive ion etching (DRIE) parameters of used recipes in this study. *R*_0_ is defined according to Equation (1).

Recipe No.	Pressure (mTorr)	Radiofrequency (RF) (W) etch/dep.	Inductively Coupled Plasma (ICP) (W) etch/dep.	*R*_0_ (nm/s)
1	15	50/50	800/800	52.4
2	25	50/50	800/800	65.3
3	25	40/40	700/700	19.3
4	25	50/50	800/800	73.3
5	20	50/50	800/800	54.0
6	20	30/20	600/600	18.0

**Table 2 micromachines-11-00864-t002:** *S* values fitted from experimental data with Equation (5).

Recipe No.	1.2 μm	2 μm	3 μm	4 μm	5.25 μm
1	0.0143	n/a	n/a	n/a	n/a
2	0.0315	n/a	n/a	n/a	n/a
3	0.0040	n/a	n/a	n/a	n/a
4	n/a	0.0194	0.0097	0.0197	0.0174
5	n/a	0.0183	0.0091	0.0146	0.0116
6	n/a	n/a	0.0047	n/a	n/a

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
