# Peer review of "Towards the Fabrication of High-Aspect-Ratio Silicon Gratings by Deep Reactive Ion Etching"

_micromachines, 2020, doi:10.3390/mi11090864_

Round 1

Reviewer 1 Report

The present manuscript presents an investigation of the Bosch process for the manufacturing of high-aspect-ratio gratings in silicon. The Bosch process is very well investigated and therefore I rate the overall novelty of the paper as low. However, the systematic investigation of etch parameters for the special case of a grating is of high interest for anybody who wants to fabricate these kind of structures. The fabrication results are of high quality and a good indication of what you can do with the Bosch process with respect to aspect ratio and grating period. Therefore I recommend publication of the manuscript.

Author Response

We thank the referee for the time spent on reviewing the draft and positive comments on our work! The motivation of this study is exactly as the referee said, we wanted to dig more into the well-investigated Bosch process, and we hope that this study could benefit the researchers who are interested in fabricating high aspect ratio silicon structure arrays with DRIE method, like ourselves.

Reviewer 2 Report

Authors have carried out comprehensive study about effect of various parameters in DRIE on resultant Si grating structures. The results are presented well and there are a few minor questions.

1) Patterns are made on entire 4 inch surface. How good is the uniformity of the gratings in terms of height, spacing, and others in entire 4 inch area?

2) What is the recommended size of the gratings for X-ray applications? 

3) Grating periods are around 1 ~ 10 um. Do authors think your approach (DRIE) will work on sub-micron region? Please provide your reason for your input.

4) Can you benchmark your result with other current state-of-art technique? Table will be a good format for a clear comparison.

5) Do authors think metal-assisted chemical etching can make good gratings with similar dimensions? If so, what is advantage of DRIE? If not, what makes metal-assisted chemical etching not suitable for making the same structures with the same dimensions? 

Author Response

We thank the referee for the valuable comments, which are very helpful for improving the quality of our paper and generating ideas for future work. Please, find our specific comments to your questions below.

1) Patterns are made on entire 4 inch surface. How good is the uniformity of the gratings in terms of height, spacing, and others in entire 4 inch area?

We do have some slight differences in the height of the trenches, if comparing the edge area with the central area on the wafer. It relates to the period of the grating: smaller period, larger difference (period 1.2 μm: < 2.5%; period 2 μm: < 1.5%). But these differences are acceptable for our applications, and we can’t cleave and check all the wafers (gratings need to remain in one piece for experiments), therefore we didn’t discuss about it in the paper. Besides, we didn’t see significant difference in other features that concern us most, such as duty cycle, sidewall angle, and etc.

We also have some local height variance issue, but it’s lithography induced, and will be discussed in other paper.

2) What is the recommended size of the gratings for X-ray applications?

The sizes of the gratings depend highly on the design of the setups (type of light source and size of detector). For setups using synchrotron beam as light source, the sizes of the gratings are usually very small, since the beam coverage itself was not that large (20 mm x 20 mm is already good enough for some beamline experiment). For setups with X-ray tube as light sources, we need to fabricate full 4-inch size gratings to cover the field of view of the photon detector. Now we are working on setups with even larger field of view for clinic uses. For that purpose, we are developing process for the fabrication of 8-inch gratings.

3) Grating periods are around 1 ~ 10 um. Do authors think your approach (DRIE) will work on sub-micron region? Please provide your reason for your input.

We trust MACE and cyro-etching more for the fabrication of gratings with submicron periods. Due to the nature of the Bosch process, it gives scallops to the sidewall of the etched gratings, which you may see in Fig 2b from the draft. The widths of the scallops are usually around few tens of nanometers. The scallop defect will introduce an error to the averaged duty cycle of the gratings. For gratings with large periods, the scallop defects are considered as minor issues, and could be ignored. But for gratings with submicron periods, since the size of the scallops are already comparable to the width of the grating lines, it will no longer be acceptable. For submicron applications which are not so strict with profile control or even some roughness on the surface are preferred (surface catalysis, etc.), Bosch process can still be considered as a feasible candidate.

4) Can you benchmark your result with other current state-of-art technique? Table will be a good format for a clear comparison.

Indeed, a table will be ideal to demonstrate the state of the art. However, it’ll be too complexed to make direct comparisons between different techniques, since the etching behaviors varies a lot depending on the method, notwithstanding the fact that there are some particular requirements on the etched features for X-ray grating interferometry applications. Besides, to make the comparison quantitative, performance measured under X-ray is also indispensable, which is hardly possible for now. But currently we are working on multiple approaches of grating fabrication, and we hope we can make a comprehensive comparison as suggested here, in future publications.

5) Do authors think metal-assisted chemical etching can make good gratings with similar dimensions? If so, what is advantage of DRIE? If not, what makes metal-assisted chemical etching not suitable for making the same structures with the same dimensions? 

One of the co-authors actually works on metal-assisted chemical etching, and we are fabricating gratings with submicron periods or exotic patterns using MACE method. Actually we already see some impressive results from MACE recently. However, we still have some pending issues with MACE: the process is not as robust as Bosch yet, especially on large areas. With Bosch etching, we can control the profile of the grating for some special purposes, which are hard to achieve with MACE. Therefore, we will continue working on both approach (wet and dry).

Inspired by the questions, we made some adjustment to our draft:

  1. It’s a very good suggestion that we should let the readers understand why we want to fabricate large area gratings with good uniformity, so we implemented the answers to question 1/2 to the draft (from line 250 to line 255 on page 8; from line 63 to line 64 on page 2).
  2. Making a table comparing different micro- / nano- fabrication techniques is not easy to realize, however we provided the readers with some of the recently reported results as benchmark.